

# Technical note: The beneficial role of stratigraphy on slope stabilization by drainage trenches

Gianfranco Urciuoli[1], Luca Comegna[2], Marianna Pirone[1], Luciano Picarelli[2]

[1]Dipartimento di Ingegneria Civile, Edile e Ambientale, Università di Napoli "Federico II", Napoli, 80125, Italy
[2]Dipartimento di Ingegneria, Università della Campania "Luigi Vanvitelli", Aversa, 81031, Italy

*Correspondence to*: Luca Comegna (luca.comegna@unicampania.it)

**Abstract.** Slope stabilization through drainage trenches is a classic approach in geotechnical engineering. Considering the low hydraulic conductivity of the soils in which this measure is usually adopted, a major constraint to the use of trenches is

the time required to obtain a significant pore pressure decrease, here called "time lag". In fact, especially when the slope safety factor is small, the use of drainage trenches may be a chancy approach due to the probability that slope deformations will damage the system well before it will become fully operative.

However, this paper shows that the presence of persistent pervious natural soil layers in the slope can provide a significant benefit by increasing drainage efficiency and reducing time lag. As a matter of fact, any pervious layer that is intercepted by

trenches may operate as part of the global hydraulic system, reducing the drainage paths.

A simplified approach to design a drainage system accounting for the presence of a persistent pervious layer is proposed. This approach, which can exploit solutions available in literature for parallel drainage trenches, has been validated by numerical analyses.

## 1 Introduction

The stabilization of deep landslides in clay is one of the greatest challenges to engineers due to the high cost and the unreliability of many structural solutions. Often, the only available approach is by deep drainage, which can lead to some shear strength increase through a generalized pore pressure decrease. Available solutions (Hutchinson, 1977; Bromhead, 1984; Stanic, 1985; Desideri et al., 1997; Pun and Urciuoli, 2008; Urciuoli and Pirone, 2013) concern the case of deep parallel trenches (and of deeper drainage panels as well), which is dealt with also in this paper, and the case of tubular drains

in a homogeneous soil.

Considering the fine-grained nature of the soil, a major constraint to slope stabilization by draining trenches is the long time required to obtain a significant pore pressure decrease (time lag). Especially when the slope is characterised by a small safety factor or is subjected to slow movements, the use of draining trenches is in fact problematic due to the probability that slope deformations will damage the system well before it will become fully operative thus vanishing its potential effectiveness.

However, as higher is the depth of trenches (or of drainage panels) as higher the probability that these intercept even thin soil layers of higher hydraulic conductivity at an intermediate depth between the ground surface and the slip surface. This would





be a lucky chance since the incorporation of such layers in the drainage system may play a highly beneficial role on both the time to attain the final steady-state condition, and the system efficiency.

The scope of this paper is just examining the influence on the drainage system, of a pervious soil layer parallel to the ground

surface.

## 2 The basic model

The solutions presented below are based on the following assumptions:

-        the groundwater flow is two-dimensional;

-        each soil layer is homogeneous, isotropic and is characterized by a linear elastic constitutive law;

-        total stresses are constant during the consolidation process (this allows to uncouple the analysis of the hydraulic and of the mechanical soil response).

The governing equation of the problem (i.e. soil consolidation induced by the draining elements) is the following:

$$h_t - c_v^{2D}(h_{xx} + h_{yy}) = 0 \ , \tag{1}$$

where $h = \zeta + \frac{u}{\gamma_w}$ and $c_v^{2D} = \frac{KE}{2(1+\nu)(1-2\nu)\gamma_w}$.

The technical literature reports solutions concerning the case of parallel draining trenches and of tubular drains in homogeneous soils, which are generally presented in the form of dimensionless design charts, providing the average efficiency, $\bar{E}(t,\Gamma)$, along the slip surface $\Gamma$:

$$\bar{E}(t,\Gamma) = \frac{u(0,\Gamma) - \bar{u}(t,\Gamma)}{u(0,\Gamma)} \ . \tag{2}$$

In Eq. (2), $u(0,\Gamma)$ is the initial pore pressure on the slip surface, $\Gamma$, and $\bar{u}(t,\Gamma)$ is the average pore pressure at time $t$

modified by the draining elements; $u(0,\Gamma)$ is generally assumed to be hydrostatic. During the consolidation phase, pore pressures decrease towards the minimum steady-state value $\bar{u}(\infty,\Gamma)$, which is attained at time $t \to \infty$ when the efficiency $\bar{E}(\infty,\Gamma)$ reaches the highest value.

The available solutions for parallel trenches, featured by a thickness $H_0$ and a width $b$, consider the soil volume between the two axes of symmetry, which respectively coincide with the middle of a trench and the centreline between two adjacent

trenches (Fig. 1a). This volume is delimited by the ground surface and an impervious bottom surface located at the distance $H$ from the ground surface. The ground and bottom surfaces are both horizontal: the slope angle is indeed assumed to play a negligible role on the hydraulic process (Aloi et al., 2019). The slip surface $\Gamma$ is a horizontal plane as well, located at depth $D$. In this paper it is assumed to be coincident with the base of trenches ($D = H_0$).

A key hypothesis, which strongly affects the solution, is the presence of a permanent film of water at the ground surface

(Burghignoli and Desideri, 1987; D'Acunto and Urciuoli, 2006; D'Acunto et al., 2007; D'Acunto and Urciuoli, 2010). However, due to local formation of water ponding and saturation of vertical cracks in the ground, often this is not far from the truth, at least during the wet season. Based on this assumption, the pore pressure decrease is uniquely due to rotation of

the flow lines towards the drainage trenches. Pore pressures in the zone between parallel trenches are then at any time less than hydrostatic (Fig. 1b). In contrast, beyond the bottom of trenches, the upward direction of the flow lines leads to a pore

pressure distribution higher than hydrostatic. It is just for this reason that the drains should always reach a depth close to the slip surface.

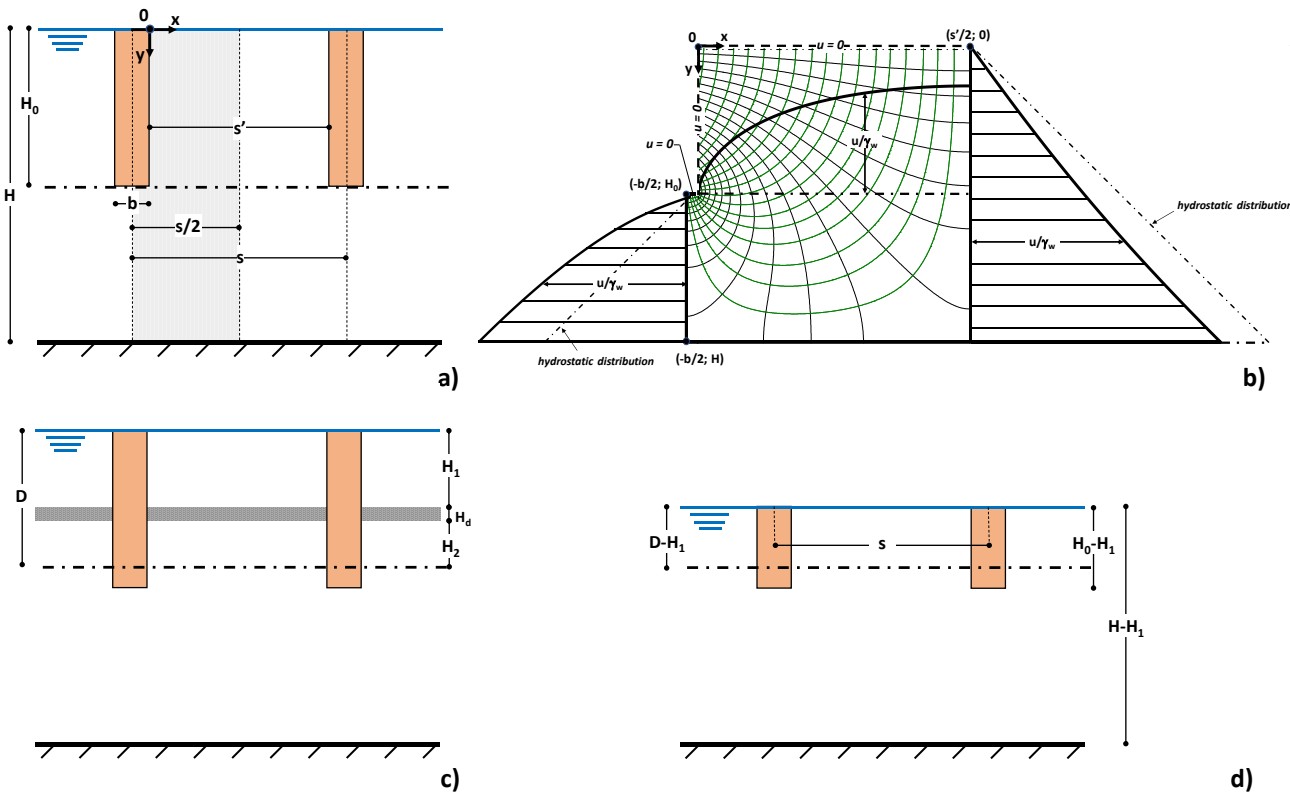

Figure 1: (a) Schematic representation of the case at hand. (b) Flow lines and equipotential lines in homogeneous soil; piezometric heads along the vertical axes at the middle of the trench, at the centreline between two adjacent trenches and on the horizontal

plane at depth of trench bottom. (c) Scheme with an intermediate pervious layer; (d) equivalent scheme with a water film at the depth of the uppermost boundary of layer $d$ .

## 3 Influence of a pervious layer located at an intermediate depth between ground and slip surface

### 3.1 Time of consolidation

As outlined above, the presence of one or more persistent pervious layers in the soil body to be stabilized (a not unlikely

situation in deep clay deposits to be stabilised with draining panels) may play a highly beneficial role on time lag and effectiveness of the drainage system.





The influence of a layer parallel to the ground surface, here indicated as the "draining layer $d$ ", featured by a thickness $H_d$ as in Fig. 1c, has been investigated by FEM analyses using the code SEEP® (GEO-SLOPE Int. Ltd., 2012). The cases examined in this paper are indicated in Tab. 1; the results of the analyses have been elaborated in a dimensionless form.


**Table 1: Examined cases studied with FEM.**

| Numerical analyses by SEEP/W | | Soil properties | | | | | Geometry | | | |
|---|---|---|---|---|---|---|---|---|---|---|
| | | $K$ (m/s) | $K_d/K$ - | $\theta$ - | $E$ (kPa) | $\nu$ - | $H_0$ (m) | $s/H_0$ - | $b/H_0$ - | $H/H_0$ - |
| Homogenous soil model $H_d = 0$ | | $10^{-7}, 10^{-9}$ | - | 0.5 | 15000 | 0.3 | 10, 20, 30 | 1,2,3 4,5,6 | 0.16 | 1,1.5, 2.5 |
| | | | | | | | | | 0.16 | 1 |
| Stratified soil model $H_d = 0.025\ H_0$ | $H_1 = 0.25\ H_0$ | $10^{-7}, 10^{-9}$ | 10,100, 1000,10000 | 0.5 | 15000 | 0.3 | 10,20,30 | 1,2,3 4,5,6 | 0.16 | 1 |
| | $H_1 = 0.50\ H_0$ | $10^{-7}, 10^{-9}$ | 10,100, 1000,10000 | 0.5 | 15000 | 0.3 | 10,20,30 | 1,2,3 4,5,6 | 0.16 | 1,1.5, 2.5 |
| | $H_1 = 0.75\ H_0$ | $10^{-7}, 10^{-9}$ | 10,100, 1000,10000 | 0.5 | 15000 | 0.3 | 10,20,30 | 1,2,3 4,5,6 | 0.16 | 1 |

The data show that the presence of the pervious layer allows to significantly shorten the time lag, here represented by the
time factor, $T_{90}$, corresponding to an efficiency $\bar{E}(t_{90}, \Gamma) = 90\%$:

$$T_{90} = \frac{c_v^{2D} t_{90}}{H_0^2} \ . \tag{3}$$

Figures 2a and 2b, which report some results concerning the horizontal plane located at depth $D = H_0$, suggest quite a rapid attainment of $\bar{E} = 90\%$, which is a crucial issue of the design. For significant values of trench spacing in the practice (i.e. $s/H_0 < 3$), the following considerations may be drawn: i) for $H_1/H_0 = 0.75$ and $K_d/K = 100$ (Fig. 2a), the dimensionless time
$T_{90}$ ranges between one half and one third of the value that would be obtained in the absence of the draining layer; ii) for





$K_d/K = 1000$ (Fig. 2b), $T_{90}$ significantly decreases with depth of the layer $d$ (for $H_1/H_0 = 0.75$, it drops to about 20% of the value obtainable in homogenous soils).



**Figure 2: Results of the FEM analyses (assuming $H = H_0$). Dimensionless time, $T_{90}$, as a function of trench spacing and of (a) $K_d/K$ ratio for $H_1/H_0 = 0.75$ and (b) $H_1/H_0$ ratio for $K_d/K = 1000$. Average steady-state efficiency as a function of trench spacing and (c) $K_d/K$ ratio for $H_1/H_0 = 0.75$; (d) $H_1/H_0$ ratio for $K_d/K = 1000$. (e) Dimensionless pore pressure over the lowermost boundary of layer $d$, $u_d/(\gamma_w H_d)$, as a function of $x/(s'/2)$ and $s/H_0$, for $K_d/K = 1000$ and $H_1/H_0 = 0.75$. (f) Values of $s_d/H_0$ as a function of $K_d/K$ and $H_1/H_0$.**





## 3.2 Steady state condition

The presence of a pervious layer allows to obtain higher values of $\bar{E}(\infty, \Gamma)$, and sooner than in homogeneous soils. Some significant data are provided:

i)    in Fig. 2c, where the steady-state efficiency for $H_1/H_0 = 0.75$ is reported as a function of the ratio $K_d/K$ and of trench spacing; as shown, as higher is the hydraulic conductivity of the draining layer as higher the efficiency (as an example, for $K_d/K = 1000$ and $s/H_0 = 3$ it practically doubles); a major effect of layer $d$ is in fact

diversion of a significant part of water coming from the ground surface towards the trench thus strongly reducing water towards the slip surface;

ii)    in Fig. 2d, where the efficiency for $K_d/K = 1000$ is reported as a function of depth of layer $d$ and trench spacing; the figure shows that it increases as the dimensionless distance, $H_1/H_0$, increases; the effect of layer $d$ is a strong pore pressure reduction at depth $H_1$; as a consequence, pore pressure decrease, due to the action of layer

$d$, increases with its depth;

iii)    in Fig. 2e, where, the non-dimensional pore pressure distribution, $u_d$, along the lower boundary of the draining layer $d$ is plotted as a function of trench spacing for $H_1/H_0 = 0.75$ and $K_d/K = 1000$; near the trench boundary, the pressure head is less than $H_d$, hence, a free water surface forms in the layer $d$ (here water can move towards the trench only below this surface, where pore pressures are positive).

### 3.2.1 A simplified approach to predict the steady-state condition

In the following, a simplified model for the optimization of the design is briefly described. A very efficient working condition is achieved if, at the centreline between two adjacent trenches, the atmospheric pressure is attained at the uppermost point of layer $d$. Any further increment of the water level would generate an artesian condition. This is what is here called "full draining layer activation".

The first step in the design of the drainage system is just creating the conditions for full layer activation. This is obtained when the spacing, $s$, of trenches is equal to the value $s_d$, according to the following expression:

$$\frac{s_d}{H_0} = f\left(\frac{H_d}{H_0}, \frac{H_1}{H_0}, \frac{K_d}{K}\right) .$$ (4)

Numerical values of $s_d/H_0$ are reported in Fig. 2f.

In case of full activation of layer $d$, the response of the entire draining system may be analysed by a simplified approach.

Since the fluid pressure at the uppermost boundary of the layer $d$ is equal to the atmospheric pressure (or to a small suction especially near the trench boundary), a water film may be fictitiously assumed at the same depth (Fig. 1d). This obviously leads to a generalized pore pressure decrease in the lowermost soil. In the following, any parameter referred to this fictitious condition will be indicated with the apex*.





**Table 2: Comparison among steady-state efficiency values computed by Eq. (8), $\overline{E}(\infty)$, and FEM analyses, $\overline{E}(\infty)_{FEM}$ .**
$(\Delta = \left|\frac{\overline{E}(\infty)_{FEM} - \overline{E}(\infty)}{\overline{E}(\infty)_{FEM}}\right|\%)$. **Only results obtained for fully drained activation are reported ($s \le s_d$).**

| $H_1/H_0$=0.50 | $\overline{E}(\infty)$ | $\overline{E}(\infty)_{FEM}$ | | | | | | | |
|---|---|---|---|---|---|---|---|---|---|
| | | $K_d/K$=10 | $\Delta$ % | $K_d/K$=100 | $\Delta$ % | $K_d/K$=1000 | $\Delta$ % | $K_d/K$=10000 | $\Delta$ % |
| $s/H_0$=1 | 0.766 | | | 0.780 | 1.73 | 0.783 | 2.10 | 0.783 | 2.10 |
| $s/H_0$=1.5 | 0.697 | | | | | 0.697 | 0.00 | 0.699 | 0.28 |
| $s/H_0$=2 | 0.642 | | | | | 0.646 | 0.54 | 0.651 | 1.30 |
| $s/H_0$=3 | 0.594 | Draining layer not fully activated | | | | | | 0.601 | 1.08 |
| $s/H_0$=4 | 0.570 | | | | | | | 0.576 | 0.95 |
| $s/H_0$=5 | 0.556 | | | | | | | 0.559 | 0.53 |
| $s/H_0$=6 | 0.546 | | | | | | | | |

| $H_1/H_0$=0.75 | $\overline{E}(\infty)$ | $\overline{E}(\infty)_{FEM}$ | | | | | | | |
|---|---|---|---|---|---|---|---|---|---|
| | | $K_d/K$=10 | $\Delta$ % | $K_d/K$=100 | $\Delta$ % | $K_d/K$=1000 | $\Delta$ % | $K_d/K$=10000 | $\Delta$ % |
| $s/H_0$=1 | 0.821 | | | | | 0.833 | 1.41 | 0.834 | 1.50 |
| $s/H_0$=1.5 | 0.797 | | | | | 0.803 | 0.71 | 0.807 | 1.20 |
| $s/H_0$=2 | 0.785 | | | | | | | 0.795 | 1.22 |
| $s/H_0$=3 | 0.773 | Draining layer not fully activated | | | | | | 0.782 | 1.11 |
| $s/H_0$=4 | 0.767 | | | | | | | 0.774 | 0.83 |
| $s/H_0$=5 | 0.764 | | | | | | | 0.768 | 0.52 |
| $s/H_0$=6 | 0.761 | | | | | | | | |

The values of $\bar{u}^*(\infty, \Gamma)$ and $\overline{E}^*(\infty, \Gamma)$ may be calculated from the scheme of Fig. 1d, as a function of spacing, through the solutions for parallel trenches in homogeneous soil. The steady-state efficiency is

$$E^*(\infty, \Gamma) = \frac{u^*(0, \Gamma) - \bar{u}^*(\infty, \Gamma)}{u^*(0, \Gamma)},$$ (5)

thus

$$\bar{u}^*(\infty, \Gamma) = u^*(0, \Gamma)\left(1 - E^*(\infty, \Gamma)\right) = \gamma_w(D - H_1)\left(1 - E^*(\infty, \Gamma)\right).$$ (6)

It is worth to notice that

$$\bar{u}(\infty, \Gamma) = \bar{u}^*(\infty, \Gamma)$$ (7)

and





$$\bar{E}(\infty, \Gamma) = \frac{u(0,\Gamma) - \bar{u}^*(\infty,\Gamma)}{u(0,\Gamma)} \ .$$ (8)

The values calculated with Eq. (8) have been compared to those obtained by FEM (Tab. 2). The good agreement allows validating the proposed method. It is worth to mention that the solid lines in Figs. 2c and 2d for $H_d = 0$ are just those that are reported in the design charts.

## 4 Conclusions and final considerations

The scope of this paper is to demonstrate that the presence of soil layers of higher permeability, a not unlikely condition in
some deep landslides in clay, may be exploited to improve the efficiency of systems of drainage trenches for slope stabilization. Once established the depth of trenches, which should reach the slip surface, the selection of a proper spacing may create a hydraulic system in which such layers can work as additional drains. The problem has been examined for the case that a unique pervious layer is present at an elevation higher than the bottom of trenches.

The results of numerical analyses show that it significantly speeds up the consolidation process triggered by drainages,
leading also to a higher steady efficiency of the system. However, as mentioned in the Introduction, in many practical cases the critical aspect of the design concerns the time requested to achieve an adequate effective stress and safety factor increase. In these cases, trench spacing should be established looking essentially at the $T_{90}$ value.

If pore pressures in the draining layer do not exceed the atmospheric pressure, a hydraulic disconnection forms between the two parts of the landslide body respectively located above and below the layer. In such a way, the water film which is
normally assumed at the ground surface ideally moves to the depth of the draining layer. This simple consideration allows to employ the design charts available for the design of drainage trenches in homogeneous soils in the equivalent scheme characterised by groundwater level located at the depth of the draining layer, in order to calculate the final system efficiency. It is worth to mention that the hydraulic continuity of layer $d$ is a fundamental condition for the design. Considering the variability and the unpredictability of many natural situations, proper investigations including the observational method to
check the validity of such an assumption, are then warmly recommended.

## Acknowledgments

This research has been developed within the framework of the PRIN 2015 project titled "Innovative Monitoring and Design Strategies for Sustainable Landslide Risk Mitigation", funded by the Italian Ministry of Education, University and Research (MIUR).



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

**List of symbols**

| | |
|---|---|
| $b$ | width of the trench |
| $c_v^{2D}$ | 2D coefficient of consolidation |
| $d$ | draining layer |
| $D$ | depth of the slip surface |





| | |
|---|---|
| $E$ | Young modulus of the soil |
| $\bar{E}(t, \Gamma)$ | average efficiency of the draining trenches at time $t$ along the sliding surface $\Gamma$ |
| $\bar{E}(\infty, \Gamma)$ | average steady-state efficiency of the draining trenches along the sliding surface $\Gamma$ |
| $\bar{E}^*(\infty, \Gamma)$ | average steady-state efficiency of the draining trenches along the sliding surface $\Gamma$ according to the simplified approach (full activation of layer $d$) |
| $\gamma_w$ | unit weight of water |
| $\Gamma$ | slip surface |
| $h$ | total head |
| $h_t$ | first derivative of total head $h$ with respect to time $t$ |
| $h_{xx}$ | second derivative of total head $h$ with respect to abscissa $x$ |
| $h_{yy}$ | second derivative of total head $h$ with respect to ordinate $y$ |
| $H$ | depth of the impervious bottom surface |
| $H_0$ | depth of the base of trench |
| $H_1$ | depth of the draining layer $d$ |
| $H_d$ | thickness of the draining layer $d$ |
| $K$ | coefficient of hydraulic conductivity |
| $K_d$ | coefficient of hydraulic conductivity of the draining layer $d$ |
| $\theta$ | soil moisture |
| $s$ | spacing between trenches |
| $s_d$ | spacing between trenches creating the conditions for full activation of the draining layer $d$ |
| $s'$ | distance between the boundaries of the trenches |
| $v$ | Poisson ratio of the soil |
| $t$ | time |
| $t_{90}$ | dimensional time corresponding to $\bar{E}(t, \Gamma) = 90\%$ |
| $T_{90}$ | dimensionless time factor for $\bar{E}(t, \Gamma) = 90\%$ |
| $u$ | pore pressure |
| $u_d$ | pore pressure at the base of the draining layer $d$ |
| $u(0, \Gamma)$ | initial pore pressure (time $t=0$) on the slip surface $\Gamma$ |
| $u^*(0, \Gamma)$ | initial pore pressure (time $t=0$) on the slip surface $\Gamma$ according to the simplified approach (full activation of the draining layer $d$) |
| $\bar{u}(t, \Gamma)$ | average pore pressure at time $t$ on the slip surface $\Gamma$, modified by drainage trenches |
| $\bar{u}(\infty, \Gamma)$ | average steady-state pore pressure on the slip surface $\Gamma$, modified by drainage trenches |



| $\bar{u}^*(\infty, \Gamma)$ | average steady-state pore pressure on the slip surface $\Gamma$, modified by drainage trenches according to the simplified approach (full activation of the draining layer $d$) |
| $\zeta$ | elevation head |