# Peer review of "Technical note: The beneficial role of a natural permeable layer on slope stabilization by drainage trenches"

_Hydrology and Earth System Sciences, 2019_

## Referee Comment (RC1) · Anonymous Referee #1 · 26 Jan 2020

The paper shows that the eventual presence of a persistent pervious natural soil layer intercepting the drainage trenches in a slope has the beneficial effect of increasing drainage efficiency and reducing time lag. The manuscript is well written with clear figures and tables. The Authors also provide the minimum trench spacing needed to attain the atmospheric pressure at the middle plane, so that a condition of water table can be assumed at the top of the pervious layer. A simplified procedure is developed to analyse this case that provides results in a good agreement with those computed by FE seepage analyses. Then, the design charts available for the design of drainage trenches in homogeneous soils can be still used referring to the equivalent scheme proposed by the Authors.

The paper can be accepted for publication in its present form.

In the following just a few notes:

(1) In Table 1, first row, last column there are two subcases, but the second seems to be included in the first one in that both refer to b/H0 =0.16 and H/H0 =1, please check;

(2) Page 6, line 111: please cancel "where";

(3) Page 6, line 118: the sentence " Any further increment of the water level would generate an artesian condition" could be eliminated in that introduces an additional comment not needed to understand the concept of reduced seepage domain.

———————————————

---

## Referee Comment (RC2) · Anonymous Referee #2 · 29 Jan 2020

The manuscript presents an interesting theoretical assessment of the effect of a highly permeable layer in a low permeable soil on drainage efficiency. I understand from the manuscript that the first part shows how t90 decreases with the presence of an permeable layer. The second part describes a simplified approach to calculate drainage efficiency. I recommend major revisions, mainly with regards to the second part, as detailed in the comments below.

1. Methodology and derivation of equations I appreciate the page limit on a technical note, but some of the equations are not clear.  - How is the term t90 in equation 3 obtained from either equation 2?  Is there a closed form solution or is this obtained

through optimisation of the numerical solution? - Can you elaborate on how you got equation 4? Is this based on the interpretation of the numerical experiment or is this derived from the governing equation? Is it possible to write equation 4 as a set of differential equations? - How do you calculate pore pressures in equations 5-8? From line 145 it appears these are not based on the FEM calculation. Is there a closed-form solution?

2. Discussion of limitations of simplified approach The simplified approach is only valid if the drainage layer is fully activated. Can you add a discussion on how to determine if this condition is satisfied?

3. Discussion on real-world applicability and field testing I would recommend a section discussing how this theoretical finding could be corroborated in a field experiment (especially the concept of fully activated drainage layer)

4. Change title I suggest to make the title more specific, for instance 'The effect of a permeable layer in a low permeable soil on soil stabilisation by drainage trenches'

5. Pervious Change 'pervious' to the more commonly used word 'permeable'

---

## Author Comment (AC1) · 30 Jan 2020

Dear Referee #1, many thanks for your comments. We really appreciated your positive assessment. The goal of the paper is exactly highlighting that the presence of even a small pervious layer could have a very beneficial effect on the performance of a drainage system, thus it should be taken into account for its design. The replies to the specific suggestions, are reported in the follow.

C1 - In Table 1, first row, last column there are two subcases, but the second seems to be included in the first one in that both refer to $b/H0 = 0.16$ and $H/H0 = 1$, please check. R1 – You are right. We will fix this mistake in Table 1.

[Figure]

C2 - Page 6, line 111: please cancel "where". R2 – The word "where" in line 111 was reported in continuity with the word "where" reported at the beginning of lines 102 and 107. Anyway, it will be replaced by "showing".

C3 - Page 6, line 118: the sentence " Any further increment of the water level would generate an artesian condition" could be eliminated in that introduces an additional comment not needed to understand the concept of reduced seepage domain. R3 – We will eliminate that sentence, because effectively redundant.

---

## Author Comment (AC2) · 2 Feb 2020

Reply to RC2
* * *
**The manuscript presents an interesting theoretical assessment of the effect of a highly permeable layer in a low permeable soil on drainage efficiency. I understand from the manuscript that the first part shows how t90 decreases with the presence of an permeable layer. The second part describes a simplified approach to calculate drainage efficiency. I recommend major revisions, mainly with regards to the second part, as detailed in the comments below.**

Dear Reviewer,

we are grateful for your comments and suggestions that give us the opportunity to improve and clarify some aspects of the paper.

In the following, a reply to every point is reported.

**C1 - METHODOLOGY AND DERIVATION OF EQUATIONS**

**C1.1 - I appreciate the page limit on a technical note, but some of the equations are not clear. - How is the term t90 in equation 3 obtained from either equation 2? Is there a closed form solution or is this obtained through optimisation of the numerical solution?**

The value of $t_{90}$ has been obtained by a **numerical integration of** Eq. (1) through the following steps:

- Calculation, as a function of time, of the total heads, *h(t,x,y,z)* thus of the pore pressures u(t,x,y,z), which turn from the value $u_0$ to the value $u_\infty$ (Fig. C1a in the Discussion);
- Representation of obtained results in a dimensionless form, through the efficiency E(t,x,y,z) at any point of the domain, and calculation of the average value of the efficiency, $\bar{E}(t,\Gamma)$, along the basal plane of trenches assumed as being the failure surface, $\Gamma$;
- Determination of the time $t_{90}$, as the instant at which is $\bar{E}(t,\Gamma)=0{,}9$ (Fig. C1b in the Discussion).

[Figure]

**Figure C1.** Pore water pressure, u, as a function of logarithmic time, Logt (a); average efficiency along the failure surface $\Gamma$, $\bar{E}(t,\Gamma)$, as a function of logarithmic time, Logt (b).
* * *
**C1.2 - Can you elaborate on how you got equation 4? Is this based on the interpretation of the numerical experiment or is this derived from the governing equation? Is it possible to write equation 4 as a set of differential equations?**

Eq. (4) in the submitted paper has been obtained from the interpretation of the results of the numerical integration of Eq. (1). These have been reported in Fig. 2f of the submitted paper, which shows the dependency of $s_d/H_0$ on $K_d/K$ and $H_1/H_0$, having fixed $H_d/H_0$.

Such an equation can be obtained also in a closed form as an algebraic expression through integration of Eq. (1) by some simplifying hypotheses. Under these hypotheses (described later), it may be demonstrated that the conditions for full layer activation are obtained when the spacing, $s$, between trenches, is equal to the value $s_d$ (or minor than it), provided by the following expression:

$$\frac{s_d}{H_0} = 2\frac{H_d}{H_0}\sqrt{\frac{K_d}{K}} \qquad\qquad (4a).$$

This may be obtained from Eq. (1) of the submitted paper, written for the layer $d$, assuming $h_t = 0$ (steady state) and $h_{yy}=0$ (the pore pressure distribution along vertical profiles is assumed to be linear). The piezometric surface in the layer $d$ is then described by means of the piezometric head, $u_d/H_d$, at the base of the layer $d$, through the parabolic equation:

$$u_d = ax^2 + bx + c \qquad\qquad (5a)$$

in which the abscissa $x=0$ corresponds to the lateral face of the trench (Fig. 1a of the submitted paper). The parabolic surface represented by Eq. (5a) separates the upper part of the layer $d$, where pore pressures are negative, from the lower one, where pore pressures are positive.

The coefficients $a$, $b$ and $c$ may be obtained based on the following hydraulic boundary conditions:

- for $x = 0 \rightarrow u_d(0) = \alpha\gamma_w H_d$ (i.e. the ordinate of surface (5a) at the trench face is a fraction, $\alpha$, of the highest value $H_d$),

assuming $s'=s-b$, for $x = \frac{s'}{2} \rightarrow u_d\left(\frac{s'}{2}\right) = \gamma_w H_d$ (this is just the condition for full layer activation: the ordinate of surface (5a) is the highest one) and $\frac{\partial u_d}{\partial x} = 0$, for symmetry.

From previous conditions, it follows:

$$x = 0 \rightarrow c = \alpha\gamma_w H_d$$

$$x = \frac{s'}{2} \rightarrow u_d\left(\frac{s'}{2}\right) = a\frac{s'^2}{4} + b\frac{s'}{2} + \alpha\gamma_w H_d = \gamma_w H_d \quad (6a)$$

$$x = \frac{s'}{2} \rightarrow \left(\frac{\partial u_d}{\partial x}\right)_{x=\frac{s'}{2}} = as' + b = 0 \rightarrow a = -\frac{b}{s'}$$
* * *
By entering the values of $a$ and $c$ in Eq. (6a), it may be obtained:

$$u_d\left(\frac{s'}{2}\right) = -\frac{b}{s'}\frac{s'^2}{4} + b\frac{s'}{2} + \alpha\gamma_w H_d = \gamma_w H_d \rightarrow b\frac{s'}{4} = \gamma_w H_d(1-\alpha) \rightarrow b = 4\gamma_w\frac{H_d}{s'}(1-\alpha)$$

It is then easy to calculate the horizontal gradient $i_x$ and the flow rate $Q_x$, though the trench face in the layer $d$:

$$(i_x)_{x=0} = \frac{1}{\gamma_w}\left(\frac{\partial u_d}{\partial x}\right)_{x=0} = \frac{1}{\gamma_w}b = 4\frac{H_d}{s'}(1-\alpha);$$

$$Q_x = K_d\alpha H_d(i_x)_{x=0} = 4K_d\frac{H_d^2}{s'}(1-\alpha)\alpha.$$

The highest value of $Q_x$ is obtained for $\alpha=1/2$:

$$(Q_x)_{max} = Q_x\left(\alpha = \frac{1}{2}\right) = K_d\frac{H_d^2}{s'}.$$

The vertical flow rates, $Q_y^s$ and $Q_y^i$, through the uppermost and the lowermost boundaries of layer $d$ depend on the gradients $i_y^s$ e $i_y^i$. The numerical analyses discussed above show that $i_y^s$ is around 1 and, on the average, $i_y^i$ is equal to 0.5.

Therefore:

$$Q_y^s = K\frac{s'}{2}i_y^s; \quad Q_y^i = K\frac{s'}{2}i_y^i \rightarrow \Delta Q_y = Q_y^s - Q_y^i = K\frac{s'}{2}(1-i_y^i).$$

Finally, from the equilibrium condition of the fluid mass it is:

$$(Q_x)_{max} = \Delta Q_y \rightarrow K_d\frac{H_d^2}{s'} = K\frac{s'}{2}(1-i_y^i) \rightarrow \left(\frac{s'}{H_d}\right)^2 = 2\frac{K_d}{K(1-i_y^i)} \qquad (7a).$$

Eq. (4a) is obtained from Eq. (7a), assuming $s'\sim s$ and $i_y^i = 0.50$.

The Authors didn't believe useful to report such a detailed series of expressions into the paper, which lead to Eq. (7a), that is valid only in the case of $h_{yy}=0$ into the permeable layer and for reasonable but arbitrary values of the gradient $i_y^i$. Moreover, Eq. (4a) does not express the dependency of $s_d/H_0$ on $H_1/H_0$ , which has been evidenced by the results of the numerical analysis, because the average value of $i_y^i$ (0.5) has been assumed (while $i_y^i$ depends also on $H_1/H_0$).
**C1.3 - How do you calculate pore pressures in equations 5-8? From line 145 it appears these are not based on the FEM calculation. Is there a closed-form solution?**

Pore pressure (and efficiency) in Eq. 5-8 (indicated with *) have been taken from well known dimensionless plots present in the literature for the design of draining trenches in homogeneous soils referring to a simplified geometric scheme accounting for the presence of the permeable layer.

**C2 - DISCUSSION OF LIMITATIONS OF SIMPLIFIED APPROACH**

**The simplified approach is only valid if the drainage layer is fully activated. Can you add a discussion on how to determine if this condition is satisfied?**

The Reviewer is right: the simplified approach can be used only when the drainage layer is fully active. To check this condition, Fig. 2f of the submitted paper may be used. Being known the depth of permeable layer ($H_1$) and its hydraulic conductivity ($K_d$), the spacing between trenches should be lower than the one provided by the curves in Fig. 2f, this in order to activate the drainage layer all over its length. However, the Author will add in the text some considerations about the use of the simplified approach, suggesting, in particular, to adopt the observational method by installing some piezometers in the permeable layer. These aspects are illustrated at point C3.

**C3 - DISCUSSION ON REAL-WORLD APPLICABILITY AND FIELD TESTING**

**I would recommend a section discussing how this theoretical finding could be corroborated in a field experiment (especially the concept of fully activated drainage layer).**

The technical note ends with the following sentence (Lines 163-165) "*Considering the variability and the unpredictability of many natural situations, proper investigations, including the observational method, to check the validity of such an assumption are then warmly recommended*".

The "observational method" is a managed monitoring process of the response of an engineering work aimed at checking the validity of the design and addressing to proper modifications due to unexpected or neglected factors. Regarding drainage trenches, the installation of piezometers is highly recommended to check in real-time the efficiency of the drainage system (especially during the critical rainy season). The piezometers should be installed both in proximity of the slip surface (near and far from the trenches) and in the permeable layer.

Such or similar considerations will be added in section 4.
***I suggest to make the title more specific, for instance 'The effect of a permeable layer in a low permeable soil on soil stabilisation by drainage trenches'***

Following your suggestion, we decided to modify the title in "The beneficial role of a natural permeable layer on slope stabilization by drainage trenches".

**C5 – PERVIOUS**

***Change 'pervious' to the more commonly used word 'permeable'***

The word "pervious" will be replaced by "permeable" in the modified version of the text.

---

## Author Response (AR1)

*COMMENTS BY EDITOR*

*Dear Authors,*

*based on the Reviewers' comment, I will send out the revised version of your paper for further review.*
*Please address them all.*

REPLY TO EDITOR

Dear Editor,

we took into account all issues raised by both the Reviewers. Please find listed below:

- replies to Referee#1;
- replies to Referee#2;
- annotated version of the manuscript, reporting the modifications.

Best regards.

**COMMENTS BY REFEREE #1**

**The paper shows that the eventual presence of a persistent pervious natural soil layer intercepting the drainage trenches in a slope has the beneficial effect of increasing drainage efficiency and reducing time lag. The manuscript is well written with clear figures and tables. The Authors also provide the minimum trench spacing needed to attain the atmospheric pressure at the middle plane, so that a condition of water table can be assumed at the top of the pervious layer. A simplified procedure is developed to analyse this case that provides results in a good agreement with those computed by FE seepage analyses. Then, the design charts available for the design of drainage trenches in homogeneous soils can be still used referring to the equivalent scheme proposed by the Authors.**

REPLIES TO REFEREE #1

Dear Referee #1,

many thanks for your comments. We really appreciated your positive assessment.

The goal of the paper is exactly highlighting that the presence of even a small pervious layer could have a very beneficial effect on the performance of a drainage system, thus it should be taken into account in the design.

The replies to the specific suggestions, are reported in the following. All the modifications will be referred to the annotated version of the manuscript. In particular:

- the eliminated parts have been reported as barred words;

- the added parts have been reported in yellow.

*C1 - In Table 1, first row, last column there are two subcases, but the second seems to be included in the first one in that both refer to b/H0 =0.16 and H/H0 =1, please check.*

R1 – You are right. We fixed this mistake in Table 1.

***C2 - Page 6, line 111: please cancel "where".***

R2 – The word "where" in line 112 of the current annotated manuscript was reported in continuity with the word "where" reported at the beginning of lines 103 and 108. Anyway, it was replaced by "showing" at all the three lines.

***C3 - Page 6, line 118: the sentence " Any further increment of the water level would generate an artesian condition" could be eliminated in that introduces an additional comment not needed to understand the concept of reduced seepage domain.***

R3 – That sentence was eliminated.

*The manuscript presents an interesting theoretical assessment of the effect of a highly permeable layer in a low permeable soil on drainage efficiency. I understand from the manuscript that the first part shows how t90 decreases with the presence of an permeable layer. The second part describes a simplified approach to calculate drainage efficiency. I recommend major revisions, mainly with regards to the second part, as detailed in the comments below.*

REPLIES TO REFEREE #2

Dear Referee #2,

we are grateful for your comments and suggestions that give us the opportunity to improve and clarify some aspects of the paper.

In the following, please find a reply to every point. All the modifications will be referred to the annotated version of the manuscript. In particular:

- the eliminated parts have been reported as barred ;

- the added parts have been reported in yellow.

**C1 - METHODOLOGY AND DERIVATION OF EQUATIONS**

*C1.1 - I appreciate the page limit on a technical note, but some of the equations are not clear. - How is the term t90 in equation 3 obtained from either equation 2? Is there a closed form solution or is this obtained through optimisation of the numerical solution?*

The value of $t_{90}$ has been obtained by a **numerical integration of** Eq. (1) through the following steps:

- calculation, as a function of time, of the total heads, $h(t,x,y,z)$ thus of the pore pressures $(t,x,y,z)$, which turn from the value $u_0$ to the value $u_\infty$ (Fig. C1a);

- representation of the results in a dimensionless form, through the efficiency $E(t,x,y,z)$ at any point of the domain, and calculation of the average value of the efficiency, $\bar{E}(t,\Gamma)$, along the basal plane of trenches assumed as being the failure surface, $\Gamma$;

- determination of the time $t_{90}$, as the instant at which is $\bar{E}(t,\Gamma)=0{,}90$ (Fig. C1b).

Regarding this point, the manuscript has been integrated at lines 87-88 of the annotated version.

[Figure]

**Figure C1.** Pore water pressure, u, as a function of the logarithm of time, Logt (a); average efficiency along the failure surface Γ, $\bar{E}(t,\Gamma)$, as a function of the logarithm of time, Logt (b).

*C1.2 - Can you elaborate on how you got equation 4? Is this based on the interpretation of the numerical experiment or is this derived from the governing equation? Is it possible to write equation 4 as a set of differential equations?*

Eq. (4) has been obtained from the interpretation of the results of the numerical integration of Eq. (1). These have been reported in Fig. 2f of the paper, which shows the dependency of $s_d/H_0$ on $K_d/K$ and $H_1/H_0$, having fixed $H_d/H_0$.

Such an equation may be obtained also in a closed form as an algebraic expression through integration of Eq. (1) by some simplifying hypotheses. Under these hypotheses (described later), it may be demonstrated that the conditions for full layer activation are obtained when the spacing, $s$, between trenches, is equal to the value $s_d$ (or less than it), provided by the following expression:

$$\frac{s_d}{H_0} = 2\frac{H_d}{H_0}\sqrt{\frac{K_d}{K}} \qquad\qquad (4a).$$

This may be obtained from Eq. (1) of the paper, written for the layer $d$, assuming $h_t = 0$ (steady state) and $h_{yy}=0$ (the pore pressure distribution along vertical profiles is assumed to be linear). The piezometric surface in the layer $d$ is then described by means of the piezometric head, $u_d/H_d$, at the base of the layer $d$, through the parabolic equation:

$$u_d = ax^2 + bx + c \qquad\qquad (5a)$$

in which the abscissa *x=0* corresponds to the lateral face of the trench (Fig. 1a). The parabolic surface represented by Eq. (5a) separates the upper part of the layer *d*, where pore pressures are negative, from the lower one, where pore pressures are positive.

The coefficients *a*, *b* and *c* may be obtained based on the following hydraulic boundary conditions:

- for $x = 0 \rightarrow u_d(0) = \alpha \gamma_w H_d$ (i.e. the ordinate of surface (5a) at the trench face is a fraction, $\alpha$, of the highest value $H_d$),

- assuming $s'=s-b$, for $x = \frac{s'}{2} \rightarrow u_d\left(\frac{s'}{2}\right) = \gamma_w H_d$ (this is just the condition for full layer activation: the ordinate of surface (5a) is the highest one) and $\frac{\partial u_d}{\partial x} = 0$, for symmetry.

From previous conditions, it follows:

$$x = 0 \rightarrow c = \alpha \gamma_w H_d$$

$$x = \frac{s'}{2} \rightarrow u_d\left(\frac{s'}{2}\right) = a\frac{s'^2}{4} + b\frac{s'}{2} + \alpha \gamma_w H_d = \gamma_w H_d \quad (6a)$$

$$x = \frac{s'}{2} \rightarrow \left(\frac{\partial u_d}{\partial x}\right)_{x=\frac{s'}{2}} = as' + b = 0 \rightarrow a = -\frac{b}{s'}$$

By entering the values of *a* and *c* in Eq. (6a), it may be obtained:

$$u_d\left(\frac{s'}{2}\right) = -\frac{b}{s'}\frac{s'^2}{4} + b\frac{s'}{2} + \alpha \gamma_w H_d = \gamma_w H_d \rightarrow b\frac{s'}{4} = \gamma_w H_d(1 - \alpha) \rightarrow b = 4\gamma_w \frac{H_d}{s'}(1 - \alpha)$$

It is then easy to calculate the horizontal gradient $i_x$ and the flow rate $Q_x$, though the trench face in the layer *d*:

$$(i_x)_{x=0} = \frac{1}{\gamma_w}\left(\frac{\partial u_d}{\partial x}\right)_{x=0} = \frac{1}{\gamma_w}b = 4\frac{H_d}{s'}(1 - \alpha);$$

$$Q_x = K_d \alpha H_d(i_x)_{x=0} = 4K_d \frac{H_d^2}{s'}(1 - \alpha)\alpha.$$

The highest value of $Q_x$ is obtained for $\alpha=1/2$:

$$(Q_x)_{max} = Q_x\left(\alpha = \frac{1}{2}\right) = K_d\frac{H_d^2}{s'}.$$

The vertical flow rates, $Q_y^s$ and $Q_y^i$, through the uppermost and the lowermost boundaries of the layer $d$, depend on the gradients $i_y^s$ e $i_y^i$. The numerical analyses discussed above show that $i_y^s$ is around 1 and, on the average, $i_y^i$ is equal to 0.5.

Therefore:

$$Q_y^s = K\frac{s'}{2}i_y^s;\ Q_y^i = K\frac{s'}{2}i_y^i \rightarrow \Delta Q_y = Q_y^s - Q_y^i = K\frac{s'}{2}\left(1 - i_y^i\right).$$

Finally, from the equilibrium condition of the fluid mass, it is:

$$(Q_x)_{max} = \Delta Q_y \rightarrow K_d\frac{H_d^2}{s'} = K\frac{s'}{2}\left(1 - i_y^i\right) \rightarrow \left(\frac{s'}{H_d}\right)^2 = 2\frac{K_d}{K\left(1-i_y^i\right)} \qquad (7a).$$

Eq. (4a) is obtained from Eq. (7a), assuming s'~s and $i_y^i = 0.50$.

Accounting for the type of paper at hand (Technical Note), the Authors did not consider suitable to report such a detailed series of expressions, which lead to Eq. (7a) that is valid only in the case $h_{yy}=0$ into the permeable layer and for reasonable but arbitrary values of the gradient $i_y^i$. Moreover, Eq. (4a) does not highlight the dependency of $s_d/H_0$ on $H_1/H_0$, which has been evidenced by the results of the numerical analyses.

Please find some changes in the text at lines 124-125 of the annotated manuscript.

***C1.3 - How do you calculate pore pressures in equations 5-8? From line 145 it appears these are not based on the FEM calculation. Is there a closed-form solution?***

Pore pressure (and efficiency) in Eq. 5-8 (indicated with *) have been obtained from well known dimensionless solutions present in the literature for the design of draining trenches in homogeneous soils referring to a simplified geometric scheme, which accounts for the presence of the permeable layer. Some changes in the text may be found at lines 138-140 and at line 150 of the annotated manuscript.

**C2 - DISCUSSION OF LIMITATIONS OF SIMPLIFIED APPROACH**

*The simplified approach is only valid if the drainage layer is fully activated. Can you add a discussion on how to determine if this condition is satisfied?*

The Reviewer is right: the simplified approach may be used only when the drainage layer is fully active. To check this condition, Fig. 2f of the paper may be used. Being known the depth of permeable layer ($H_1$) and its hydraulic conductivity ($K_d$), the spacing between trenches should be lower than the one provided by the curves in Fig. 2f, this in order to activate the drainage layer all over its length.

However, the Author added in the text some considerations about the use of the simplified approach, suggesting, in particular, to adopt the "observational method" by installing some piezometers in the permeable layer or close to it. These aspects are illustrated at lines 170-176 of the annotated version of the manuscript.

**C3 - DISCUSSION ON REAL-WORLD APPLICABILITY AND FIELD TESTING**

*I would recommend a section discussing how this theoretical finding could be corroborated in a field experiment (especially the concept of fully activated drainage layer).*

We added some considerations at lines 170-176 of the annotated version.

**C4 – CHANGE TITLE**

*I suggest to make the title more specific, for instance 'The effect of a permeable layer in a low permeable soil on soil stabilisation by drainage trenches'*

Following your suggestion, we decided to modify the title in "The beneficial role of a natural permeable layer on slope stabilization by drainage trenches".

**C5 – PERVIOUS**

*Change 'pervious' to the more commonly used word 'permeable'*

The word "pervious" was replaced by "permeable" in the modified version of the manuscript.

[revised manuscript text omitted]